# Potential of Gum Arabic Functionalized Iron Hydroxide Nanoparticles Embedded Cellulose Paper for Packaging of Paneer

**DOI:** 10.3390/nano11051308

**Published:** 2021-05-15

**Authors:** Prince Chawla, Agnieszka Najda, Aarti Bains, Renata Nurzyńska-Wierdak, Ravinder Kaushik, Mansuri M. Tosif

**Affiliations:** 1Department of Food Technology and Nutrition, Lovely Professional University, Phagwara, Punjab 144411, India; princefoodtech@gmail.com (P.C.); tosifmansuri444@gmail.com (M.M.T.); 2Department of Vegetable Crops and Medicinal Plants, University of Life Science in Lublin, Doświadczalna Street 51A, 20-280 Lublin, Poland; renata.nurzynska@up.lublin.pl; 3Department of Biotechnology, Chandigarh Group of Colleges Landran, Mohali, Punjab 140307, India; aarti05888@gmail.com; 4Department of Food Technology, School of Health Sciences, University of Petroleum and Energy Studies, Dehradun, Uttarakhand 248007, India

**Keywords:** iron, cellulose, paneer, packaging, shelf-life, nanoparticles

## Abstract

Recently, the interest of scientists has turned towards eco-friendly metal nanoparticles due to their distinctive physicochemical properties that have been used in several biochemical and food applications, including drug and bioactive component delivery, sensing of food pathogenic bacteria, imaging techniques, and theranostics. Therefore, this study aimed to fabricate gum arabic stabilized iron hydroxide nanoparticles (IHNPs) using the co-precipitation process and to develop nanoparticles decorated antimicrobial cellulose paper. The agglomeration of IHNPs is a major concern, therefore, the varied concentration (0.25–2.0%) of gum arabic was used to functionalize and stabilize the nanoparticles, and based on UV-visible spectroscopy and particle size analysis, 1% gum arabic concentration was screened out. Scanning electron microscopy displayed polygonal disc shapes of IHNPs that had sides of approximately equal lengths. Energy dispersive spectroscopy was used to determine the purity of the IHNPs and results illustrated the elemental iron peak at 0.8 keV and 6.34 keV. For thermal stability, differential scanning calorimetry (DSC) was employed, and the glass transition temperature was observed at 138.50 °C with 138.31 °C onset and 147.14 °C endset temperature, respectively. Functionalized IHNPs showed a significantly (*p* < 0.05) higher zone of inhibition against *S. aureus* (29.63 mm) than that of *E. coli* and were found to be non-toxic to Caco-2 cells during cell viability assay. Time-kill kinetics showed that cellulose paper embedded with nanoparticles possessed excellent antibacterial activity against *S. aureus*. To explore the food application of developed cellulose paper, citric acid coagulated dairy product (Paneer), similar to cottage cheese was formulated, and it was evaluated for its microbial shelf life. The unwrapped sample showed higher microbial load during the fourth day of the storage. However, both wrapped samples were acceptable till the 10th of storage.

## 1. Introduction

Cellulose is the most abundantly present biopolymer on earth which has a wide range of applications in many areas of life [1]. Due to higher biocompatibility, cost-effectiveness, and negligible toxicity, cellulose is effectively used in food packaging [2,3]. Cellulose is mainly found in paper, which is commonly used as a packaging material. However, it is often a source of microbial contamination. In relation to the growing interest in natural products that can perform a combined function observed in recent years, it has contributed to the development of sustainable techniques for giving cellulose paper, which, as a packaging material, can also exhibit antimicrobial properties, ensuring the microbiological and chemical safety of food products [4]. Particularly, several types of food pathogenic bacteria can have a harmful influence on human health [5]. Although there are several antibacterial materials, interestingly, scientists are turning to the eco-friendly metal nanoparticles due to their distinctive physicochemical properties that have been used in several biochemical and food applications, including drug and bioactive component delivery, sensing of food pathogenic bacteria, imaging techniques, and theranostics [5]. Moreover, researchers are formulating eco-friendly nanoparticles of various metals, such as gold, silver, copper, zinc, iron, and magnesium [6,7]. Among these eco-friendly inorganic metal nanoparticles, iron hydroxide nanoparticles have been examined to a large extent due to their magnetic properties and low toxicity [8]. Broadly, the co-precipitation technique is used for the synthesis of iron hydroxide nanoparticles, however, aggregation of iron hydroxide nanoparticles limits their potential use. Therefore, it is necessary to functionalize the iron hydroxide nanoparticles with a suitable surface modifier [5,9]. Recently, several researchers fabricated metal nanocomposites using different plant-based and natural biopolymers and revealed the least cytotoxicity and higher antimicrobial properties [10,11,12]. Among all biopolymers, gum arabic, due to its potential surface activity, is used as an effective surface modifier for the functionalization of metal nanoparticles [5]. Nanocomposites formulated with the combination of metal nanoparticles and gum arabic are highly stable and can be used in various applications [5,11,12]. Therefore, in the current investigation, we used gum arabic as a suitable surface modifier for the functionalization of iron hydroxide nanoparticles. In contrast, cellulose paper exhibits biocidal activity and there is a need to improve the functionalization of cellulose paper to enhance the antimicrobial properties, hence impregnation of metal nanoparticles in the porous fibrous network of cellulose can improve the biocidal activity of the cellulose paper [13,14]. India is a leading country in milk production and the major portion of milk produced in India is consumed in the liquid form and the rest milk is used to prepare milk-based value-added products [15]. Among all dairy products, paneer (similar to cottage cheese) is a very popular product and it can be prepared by the heat-cum-acid coagulation of standardized buffalo or cow milk [16]. It is an excellent source of calcium and milk protein and is considered good food for agrarian people. However, due to the perishable nature of the product, the shelf-life of the product is a matter of concern [17]. Various reports have been published to improve the shelf-life of the paneer using different technology and packaging material which may enhance the cost of the product [15]. In recent years, due to the increasing and ever-growing demand for paneer by varied health-conscious people, it is necessary to improve the shelf life of the paneer with cost-effective sustainable packaging material [18]. Moreover, plastic packaging material residue may impart to paneer that causes toxic effects on human health [19]. Therefore, this study was focused on the preparation of a suitable and sustainable cellulose-based packaging material to enhance the shelf-life of paneer [20]. Our current investigation was based on the following objectives: (i) synthesis, functionalization, and characterization of iron hydroxide nanoparticles (IHNPs); (ii) antimicrobial, cytotoxicity evaluation of selected IHNPs, (iii) preparation, characterization, and antimicrobial efficacy of IHNPs embedded cellulose paper, and iv) preparation of paneer and shelf-life evaluation of panner wrapped in IHNPs embedded cellulose paper.

## 2. Materials and Methods

### 2.1. Materials

Analytical reagent grade chemicals, including ferric chloride anhydrous, sulphuric acid, nitric acid, and hydrochloric acid, were procured from Loba Chemie Private Limited, Mumbai, India. Gum acacia powder, Meuller Hinton’s Agar, potato dextrose agar, Dulbecco’s Modified Eagles Medium, L-glutamine, and streptomycin (sulfate salt) were purchased from Sigma Aldrich, St. Louis, MO, USA. Ciprofloxacin, nutrient agar, nutrient broth, and streptomycin were procured from Hi-Media Private Limited, Mumbai, India. Gram-positive and Gram-negative food pathogenic microbial strains, i.e., *Staphylococcus aureus* (MTCC 3160) and *Escherichia coli* (MTCC 443), were obtained from the Microbial Type Culture Collection (MTCC), Institute of Microbial Technology, Chandigarh, India. Human colon adenocarcinoma cells (Caco-2 cells) were collected from National Centre for Cell Science, Pune, Maharashtra, India. Plastic dishes, plates, and transwell were obtained from Corning (Corning, NY, USA). Cellular grade, triple distilled water and aqua-regia washed glassware (class “A” certified) were used during research work.

### 2.2. Methods

#### 2.2.1. Synthesis of Unmodified Iron Hydroxide Nanoparticles

The co-precipitation method was used for the synthesis of unmodified iron hydroxide nanoparticles (IHNPs). Briefly, a hot solution (55 °C) of ferric chloride anhydrous (pH < 2) was taken in a 50 mL conical flask, and the pH of the solution was adjusted (pH 8) by adding 4 M sodium hydroxide (NaOH). After the addition of NaOH, co-precipitation of the iron (III) chloride occurred and 10 mM final concentration of iron was obtained. The solution of IHNs was then transferred to a centrifuge tube and centrifuged at 10,000× *g* for 10 min. Unmodified IHNPs were then washed with triple distilled water, collected on a glass petri plate, and dried in a hot air oven at 45 °C for 4 h [21].

#### 2.2.2. Synthesis of Gum Arabic Stabilized Iron Hydroxide Nanoparticles

Different concentrations (0.25%, 0.5%, 1%, 1.5%, and 2.0%) of gum arabic were used to stabilize the IHNPs. Modified IHNs were synthesized using a solution of the gum arabic simultaneously and to this, the hot iron solution (55 °C) was added from a stock iron solution to obtain a final iron concentration of 10 mM. The sample solution was adjusted to pH 8 using 4 M NaOH and was left at 27 °C for 2 h for constant monitoring for changes in pH and maintained at pH 8 under stirring. Herein, the 10 mM ratio of iron was considered and the samples were prepared as iron: gum arabic (mM: percentage). Monodispersed gum arabic modified IHNPs were transferred into glass vials for further screening and characterization.

#### 2.2.3. Characterization of Metal Nanostructures

##### UV-Visible Spectroscopy of Iron Hydroxide Nanostructures

To confirm the synthesis and surface modification with gum arabic of IHNPs, UV-Visible spectroscopy (Evolution 201, Thermo Fischer Scientific India Pvt. Ltd., Mumbai, India) was used to evaluate the surface plasmon resonance. UV-Visible spectra of all IHNPs were observed using scan mode over the range of 800–290 nm. Monodispersed samples of IHNPs were simply taken into a cuvette (3.5 mL with 3.5 cm path length) and then any variation in surface plasmon of IHNPs was observed.

##### Particle Size and Zeta Potential of Metal Nanostructures

The dynamic light scattering (DLS) technique was employed to characterize the particle size distribution of the IHNPs dispersed in water. All gum arabic stabilized IHNPs were subjected to particle size analysis for final screening. Herein, IHNPs were dispersed in deionized water followed by ultra-sonication. Then, the particle dispersal in the liquid was mediated in a computer-controlled particle size analyzer (Zetasizer Nano ZS, Malvern Instruments Ltd. Malvern, WR14 1XZ, UK). IHNPs with smaller particle size were evaluated for zeta potential.

##### Morphological Characterization and Energy-Dispersive X-ray Spectroscopy (EDS)

The homogenous shape of the selected IHNPs was confirmed using a high-resolution transmission electron microscope functioned at an accelerating voltage of 200 kV. Briefly, a drop of a monodispersed solution of selected IHNPs was directly placed on a carbon-coated copper grid and immediately dried at 27 °C. The sample was then analyzed and a minimum of 10 images was recorded at different magnifications. The surface morphology of the selected IHNPs was assessed by scanning electron microscopy (JEOLJSM-6510LV, INCA, USA). Elemental analysis and chemical characterization were performed using Energy-dispersive X-ray spectroscopy (RONTEC’s EDX system Model QuanTax 200, USA).

##### Differential Scanning Calorimetric Analysis

Thermal analysis of selected IHNPs was carried out using differential scanning calorimeter (Shimadzu DSC-50system. Shimadzu, Kyoto, Japan). Thermograms were acquired after 2 mg powdered samples were crumpled in a standard aluminum pan heated from 20 to 350 °C with a ramping of 10 °C/min under constant purging of nitrogen (20 mL/min).

##### Antimicrobial Properties of Selected Nanoparticles

Antimicrobial properties of gum arabic stabilized IHNPs were evaluated using the agar well diffusion method against Gram-positive and Gram-negative food pathogenic strains. Briefly, Mueller Hinton Agar glass plates were inoculated with both microbial strains (1 × 10^8^ cells/mL) using the spread plate method. After 30 min incubation, wells of 6 mm in the microbial strain-loaded agar plates were punched using a cork borer. A 10 mL stock solution of different concentrations (25, 50, 75, 100 µL/mL) of IHNPs was prepared and a fixed volume (100 μL/well) from each concentration was poured into the prepared wells. Sample loaded plates were then incubated at 37 °C for 24 h. To compare the antimicrobial activity of gum arabic stabilized IHNPs with a commercially available antibiotic component, chloramphenicol was used as a standard (positive control) antibacterial agent, whereas DMSO was used as a negative control.

##### Time Kill Study

Dynamic time-kill study for IHNPs was evaluated against both Gram-Positive and Gram-Negative. Herein, after 0, 12, 18, 24, 36, and 48 h of incubation, an aliquot of 0.1 mL was taken from each plate and serially diluted and spread on Muller Hinton agar plates followed by the incubation at 37 °C for 24 h. Colonies were counted manually and Log CFU was calculated by the number of dilutions [5].

#### 2.2.4. Preparation of Nanoparticles Embedded Cellulose Paper

Gum arabic stabilized IHNPs embedded cellulose paper for packaging of paneer was prepared by following the proposed method [22]. Herein, the cellulose filter paper (15 cm × 15 cm × 10 mm) was immersed in a 1000 mL beaker containing 10 mM ferric chloride solution and selected concentration of gum arabic for 2 h. To start the coprecipitation process, 4 M NaOH was added to the solution containing the cellulose filter paper. After attaining pH 8 and to achieve complete absorption of nanoparticles, samples were kept undisturbed for 2 h. To flush the unabsorbed IHNPs, the cellulose filter paper was rinsed with tripled distilled water for 2 min. Next, rinsed cellulose paper was dried in a hot air oven at 40 °C for 30 min to remove the extra water.

##### Morphological Evaluation of Prepared Cellulose Paper

To confirm the absorption and the surface morphology of the uncoated and IHNPs embedded cellulose paper was examined using scanning electron microscopy. Briefly, plane and IHNPs embedded cellulose paper were gold-coated (20 mm thickness) using an ion coater (IB-3 Ion coater, Eiko, Tokyo, Japan) at 0.05–0.07 Torr (4 min, ion current 6 mA). Micrographs were recorded for both the samples at an acceleration voltage of 15 kV under a high vacuum (9.0 × 10^−5^ Torr).

##### Antimicrobial Properties of Prepared Cellulose Paper

Antimicrobial properties of plane and IHNPs embedded cellulose paper were evaluated using the disc diffusion method against Gram-positive and Gram-negative food pathogenic strains as described in the previous section. Herein, a 6 × 6 mm^2^ piece of both cellulose papers was loaded in the microbial strain-loaded agar plates. After incubation, the zone of inhibition (mm) was measured as the antimicrobial efficacy of the cellulose papers [5].

#### 2.2.5. Formulation of Paneer

Paneer was formulated by following the proposed method [23]. Fresh raw buffalo milk was filtered and standardized at 6% fat and 9% SNF. For paneer preparation, milk was poured into a lab-scale vat and heated up to 80 °C, and kept for 30 min holding time with constant stirring. Citric acid solution (1%) was poured into the milk after cooling at 70 °C and milk was stirred till complete coagulation. Paneer was obtained after pressing and cooling the coagulum and 2 cm × 2 cm cubes were used for the analysis.

#### 2.2.6. Influence of Packaging on the Quality of Paneer

The effect of packaging material (plane cellulose paper, and IHNPs embedded cellulose paper) on the shelf life of paneer was evaluated by sensory and microbial attributes. Paneer was wrapped in both types of paper and stored at refrigerated temperature (4–7 °C) for 12 days. Then, it was evaluated for standard plate count, yeast, and mold count.

#### 2.2.7. Statistical Analysis

The standard error mean was calculated using data tool pack analysis of Microsoft Excel, 2016 (Microsoft Corp., Redmond, WA). Statistical difference in terms of significant and non-significant values was confirmed by one-way and two-way analysis of variance and comparison between means was completed by the critical difference value [24].

## 3. Results and Discussion

### 3.1. Surface Modification and Optimization of Gum Arabic Stabilized IHNPs

The IHNPs were synthesized by co-precipitation of the ferric form of the iron salts (FeCl_3_) using an alkaline aqueous medium (NaOH) at pH 8. The synthesis of the IHNPs was confirmed by the visual change in color from light yellow (color of FeCl_3_) to brown as shown in (Figure 1a). The pH of the aqueous FeCl_3_ solution was below 1 and the iron in this aqueous solution exists as aqua-ion. With an increase of pH (after the addition of NaOH) in the solution, the aqua-ion hydrolyzed to form the hydroxo-ion, and the hydroxo-ions tried to form the complex structure, and finally, iron hydroxide is produced as a precipitate [25]. Moreover, fast aggregation of unmodified IHNPs was observed at pH 8 and this fast aggregation in IHNPs was due to the weak kinetic influence of nucleation and growth of the iron crystals during the co-precipitation process. The addition of sodium hydroxide led to sufficient super-saturation in highly soluble FeCl_3_ solution, and the energy provided by the super-saturation was quite constant. Therefore, due to this weak energy kinetics, iron hydroxide molecules diffuse through the solution, and the formation of large nuclei occurred. These coalesce iron molecules transformed into small visual aggregates. Our results are well in line with the findings of [26] who observed the formulation of large nuclei during the synthesis of iron hydroxide nanoparticles in the presence of alkali medium. Furthermore, large nucleation and aggregation in IHNPs decrease the unique properties and applications of the nanostructures. However, the usage of suitable surface functionalizing and stabilizing agents can resolve this problem effectively [5] Therefore, to maintain the monodispersed nature of the IHNPs, different concentrations of gum arabic were used as a surface modifier. As well, the initial color of FeCl_3_ was light yellow and that was changed to pale yellow after the addition of the gum arabic in a hot solution of FeCl_3._ This color change was also due to the electrostatic interaction of components of gum arabic (i.e., oligosaccharides, arabinogalactan) with FeCl_3._ Furthermore, all gum arabic stabilized IHPNs showed monodisperse nature as represented in (Figure 1b–f). This could be due to gum arabic maintained the repulsive electrostatic interactions and steric hindrance between iron molecules and water. Screening of monodispersed IHNPs was done using UV-visible spectroscopy and particle size analysis. The UV-visible spectrum of IHNPs stabilized with gum arabic is represented in (Figure 2a). Herein, all the samples showed surface plasmon resonance (SPR) at 300 nm, and overlay in the spectra was observed for all the samples except 0.25% concentration of gum arabic. It can be concluded from the results that the addition of gum arabic did not affect the SPR of IHNs. This also confirms the high stability of IHNPs with the addition of gum arabic. Further screening of IHNPs was done by particle size analysis and results of particle size of IHNPs stabilized with gum arabic are represented in Figure 2b. All the samples showed significant (*p* < 0.05) differences with each other in terms of particle size. Moreover, Fe:GA (1%) showed a significantly (*p* < 0.05) smaller particle size (6.60 nm) than that of other IHNPs, whereas Fe:GA (2%) showed a significantly (*p* < 0.05) larger particle size. Apart from the monodispersed nature of all the IHNPs, Fe:GA (0.25% and 0.50%) showed significantly (*p* < 0.05) larger particle size due to the hydrogen bonding of close-packed iron molecules. Moreover, the higher concentration of gum arabic also exhibited significantly (*p* < 0.05) higher particle size and a possible reason for the larger particle size could be that gum arabic contains charged carboxylic and amine groups that can directly adsorb onto the surface of the IHNPs, and saturation of these functional groups resulted into a large particle size of IHNPs [27]. Therefore, based on visual inspection, UV-visible characterization, and particle size analysis, IHNPs stabilized with 1% gum arabic were screened out for further analysis.

### 3.2. Characterization of IHNPs

IHNPs stabilized with 1% gum arabic was subjected to zeta potential, scanning electronic microscopy, energy-dispersive X-ray spectroscopy, transmission electron microscopy, and differential scanning calorimetry. Here, the zeta potential measurements of gum arabic stabilized IHNPs were performed to correlate the bonding of gum arabic and the overall surface charge of the IHNPs (Figure 3a). Gum arabic (1%) stabilized showed a net negative surface charge (−24.4 mV) and the source and origin of the negative surface charge was most likely due to the mainstay of gum arabic, i.e., α-(1→3)-linked d-galactopyranosyl units and side chains i.e., 2–5 β-(1→3)-linked D-galactopyranosyl units [28]. In our results, carboxyl groups got into dissociated form and the protonation of amino groups resulted in a negative surface charge on IHNPs. Therefore, due to strong electrostatic interaction, IHNPs were considered to be very stable. The morphology of gum arabic stabilized IHNPs was confirmed using scanning electron microscopy and results are represented in Figure 3b. Herein, due to high sample dispersity, gum arabic stabilized IHNPs revealed polygonal discs’ shape. Although, some gum arabic stabilized IHNPs appeared in truncated polygonal prisms shape, however, majority of the gum arabic IHNPs revealed polygonal disc shapes (curvature in the circumference) that had sides of approximately equal lengths [29]. The purity of gum arabic stabilized IHNPs was confirmed using EDS and results are represented in Figure 3c. The EDS spectrum illustrated the elemental iron peak at 0.8 keV and 6.34 keV. The presence of other elemental components, such as carbon (0.3 keV), sodium (1.06 keV), and chlorine (2.8 keV and 3.0 keV), was confirmed. Hence, the EDS spectrum was free of impurities confirming the purity of the IHNPs. Another elemental composition was attributed due to the utilization of NaOH during the synthesis of IHNPs. Results are well supported by the findings of [5], who revealed the purity of gum arabic stabilized copper nanoparticles using EDS. TEM images (Figure 3d) also revealed the nucleated and monodispersed structure of IHNPs stabilized with 1% gum arabic that could be due to electrostatic and steric interaction of gum arabic with IHNPs. DSC was used for the direct assessment of the heat energy uptake that occurred in gum arabic stabilized IHNPs within a regulated increase or decrease in temperature and results are represented in Figure 3e.

The calorimetry was particularly applied to evaluate the changes in phase transitions. Herein, the glass transition temperature of the gum arabic stabilized IHNPs was observed at 138.50 °C with 138.31 °C onset and 147.14 °C endset temperature. The occurrence of glass transitions in gum arabic stabilized IHNPs was due to the occurrence of heat change with increasing temperature. A broad peak was observed at 153.41 °C with 157.50 °C onset and 170.60 °C endset temperature that was mainly attributed to the melting point of the gum arabic stabilized IHNPs. The melting point of gum arabic stabilized IHNPs was depressed to a lower value due to reduced particle size and the peculiar shape of synthesized IHNPs. Moreover, depression in the melting point of gum arabic stabilized IHNPs could be evident that nanoparticles melt at a lower temperature than large particles of the same materials. Alteration in the melting point of nanoparticles occurs due to the larger surface to volume ratio of the nanoparticles that alter their thermodynamic and thermal properties. The results of DSC are in accordance with the findings of [30] who observed a depression in the melting point of reduced-sized octahedral-shaped silver nanoparticles decorated with polyvinyl chloride polymer.

### 3.3. Antimicrobial Properties and Time-Kill Evaluation

Antimicrobial activity of gum arabic stabilized IHNPs was evaluated against both Gram-positive and Gram-negative bacteria and the zone of inhibition was observed, and the results are presented in (Figure 4a). Herein, the zone of inhibition was ranged from 26.87–29.76 mm against both microbial strains. IHNPs showed a significantly (*p* < 0.05) higher zone of inhibition against *S. aureus* (29.63 mm) whereas, against *E. coli*, IHNPs showed a significantly (*p* < 0.05) lower zone of inhibition. However, IHNPs showed a non-significant (*p* < 0.05) difference against *S. aureus* in comparison with the standard antibiotic component (29.76 mm). According to the literature, the susceptibility of IHNPs against *S. aureus* was significantly (*p* < 0.05) higher and comparable with the standard drug due to the absence of membrane-bound periplasm in the cell membrane that consists of peptidoglycans like teichoic or teichuronic acid. Furthermore, the outer cell wall of *S. aureus* formulates a thick hydrophobic porous structure that can bind a large number of proteins and lipids and the porous membrane could be a reason for the increased permeability of chemotherapeutic agents. On the other hand, *E. coli* contains lipopolysaccharides in the outer membrane that act as an effective permeability barrier to the IHNPs and standard antibiotic drugs, hence IHNPs showed the least susceptibility against *E. coli*. Time kill kinetics of IHNPs against pathogenic bacteria was performed and results are represented in (Figure 4b). IHNPs inhibited the growth of *S. aureus* strongly as compared to *E.coli*. Here, in the case of *S. aureus,* the log CFU/mL value was significantly (*p* < 0.05) reduced from 8.46 to 7.87 in 48 h, whereas, in the case of *E. coli* log CFU/mL was significantly (*p* < 0.05) reduced from 8.42 to 8.03. Statistically, IHNPs showed significantly (*p* < 0.05) lower log CFU/mL against *S. aureus* than that of *E. coli.* The inhibition in the growth of *S. aureus* was due to the permeability of the peptidoglycan layer that allows antibiotics and IHNPs to enter the cell and resulted in protein denaturation and disruption in the cell membrane. Results were well supported by the results from our previous findings [5] in which we observed similar trends for gum arabic stabilized copper nanoparticles against Gram-positive and Gram-negative microorganisms.

### 3.4. Cytotoxicity of the Nanoparticles

Cytotoxicity of IHNPs was evaluated and the results of cell viability by MTT assay are represented in (Figure 5a). Herein, IHNPs were found to be non-toxic to Caco-2 cells. Higher concentration (10 mg/mL) of IHNPs showed 97.31% cell viability. Our results were correlated well with the literature [31] who studied the cytotoxic effect of albumin-coated copper and gum arabic caped copper nanoparticles on human breast cancer cells of MDA-MB- 231 and Caco-2 cells and reported almost no toxicity of copper nanoparticles even at higher concentration. Gum arabic stabilized IHNPs did not produce free radicals through a Fenton reaction during the cell viability of Caco-2 cells. Moreover, gum arabic controlled IHNPs inhibit the redox potential that accelerates to a catalytic cycle to produce toxic components [32].

### 3.5. Characterization of IHNPs Embedded Cellulose Paper

Morphology of gum arabic stabilized IHNPs embedded cellulose paper was confirmed using scanning electron microscopy and results are represented in (Figure 5b). As earlier discussed, due to high sample dispersity, gum arabic stabilized IHNPs revealed curvature in the circumference (polygonal discs shape). In the fibrous network of cellulose paper, polygonal disc shapes were properly visible. Micrographs of FESEM of control cellulose paper revealed unevenly distributed fiber, whereas SEM micrographs of IHNPs embedded cellulose paper revealed the functionalization of the surface of cellulose with metal nanoparticles. The high surface area of the fibrous network of cellulose paper allowed higher absorption of IHNPs on the surface. Further, the distribution of IHNPs was homogenous in the three-dimensional fibrous network of cellulose filter paper. Our results were in good correlation with the findings of [14], who revealed the morphology of a metal oxide nanoparticle-impregnated cellulose foam filter used for drinking water purification.

### 3.6. Antimicrobial Properties and Time-Kill Study of IHNPs Embedded Cellulose Paper

Antimicrobial activity of gum arabic stabilized IHNPs embedded cellulose paper was evaluated against both microbial strains and the results of the zone of inhibition are represented in (Figure 6a) Herein, all the samples showed zone of inhibition against both the microbial strains, ranging from 26.77–29.76 mm. Cellulose paper embedded with IHNPs showed a significantly (*p* < 0.05) higher zone of inhibition against *S. aureus* (29.74 mm), whereas against *E. coli* (27.11 mm), cellulose paper embedded with IHNPs showed a significantly (*p* < 0.05) lower zone of inhibition. However, IHNPs embedded cellulose paper showed a non-significant (*p* < 0.05) difference against *S. aureus* as compared to the standard antibiotic component (29.76 mm). Furthermore, time-kill kinetics of IHNPs embedded cellulose paper against pathogenic bacteria was performed and results are represented in (Figure 6b). Herein, IHNPs embedded cellulose paper inhibited the growth of *S. aureus* strongly in comparison with *E. coli*. Furthermore, during time-kill kinetics of cellulose paper and IHNPs embedded cellulose paper, it was observed that both papers showed inhibition of both microbial strains proficiently. However, IHNPs embedded cellulose paper showed a significantly (*p* < 0.05) higher reduction in the growth of both microbial stains. Moreover, inhibition in the growth of *S. aureus* was significantly (*p* < 0.05) higher due to the impregnation of gum arabic stabilized IHNPs in the unevenly distributed fiber of cellulose and the interface of different pore sizes, which is well supported by the SEM results. According to various reports, cellulose, fibrous membranes coated with metal nanoparticles showed a synergistic effect in terms of antimicrobial efficacy [13].

### 3.7. Shelf-Life Study of Cellulose Paper-Wrapped Paneer

Formulated paneer was wrapped in cellulose paper and IHNPs embedded cellulose paper to evaluate the shelf-life and results are represented in Table 1. Paneer wrapped in IHNPs embedded cellulose paper showed significantly (*p* < 0.05) lower standard plate count, coliform count, yeast, and mold counts as compared to unwrapped paneer and wrapped in control cellulose paper. During storage, control cellulose paper, and IHNPs embedded cellulose paper did not show any growth in standard plate count, coliform count, yeast and mold count up to the 4th day of storage, however, IHNPs embedded cellulose paper did not show any growth of coliform count, yeast and mold up to 6th day of storage. Unwrapped paneer showed 17 × 10^4^/g CFU and 10/g CFU of coliform, yeast, and mold count on the 2nd day of the storage. During the 8th day of the storage, IHNPs embedded cellulose paper showed a 3 × 10^4^/g CFU standard plate count, 1/g CFU coliform count, 5/g yeast, and mold count which was still acceptable according to FSSAI recommended values for paneer. The unwrapped sample showed higher microbial load during the 4th day of the storage, however, both wrapped samples were acceptable till the 10th of storage. Our results were consistent with the recommended values given by the Food Safety Standard Authority of India (FSSAI) for the paneer.

## 4. Conclusions

The co-precipitation method was used for the synthesis of iron hydroxide nanoparticles and gum arabic was used as a functionalizing agent to maintain the monodisperse properties of the nanoparticles. After optimization, scanning electron microscopy displayed the polygonal disc shape of iron hydroxide nanoparticles and energy dispersive spectroscopy revealed the purity of nanoparticles. The thermal stability of IHNPs was evaluated using DSC and nanoparticles showed the lowest melting point due to the small particle size. The susceptibility of IHNPs against *S. aureus* was significantly (*p* < 0.05) higher and comparable with the standard drug. Electron microscopy demonstrated that the nanoparticles were successfully embedded into the porous network of cellulose paper. The inhibition zone, time-kill kinetics showed that cellulose paper embedded with nanoparticles possessed excellent antibacterial activity against *S. aureus*. Paneer wrapped in nanoparticles embedded cellulose paper showed a more enhanced shelf life than that of unwrapped paneer. These results suggest that cellulose paper embedded with iron hydroxide nanoparticles could represent valuable antibacterial technology for various food packaging applications, wound dressings, and other personal care products.

## Figures and Tables

**Figure 1 nanomaterials-11-01308-f001:**
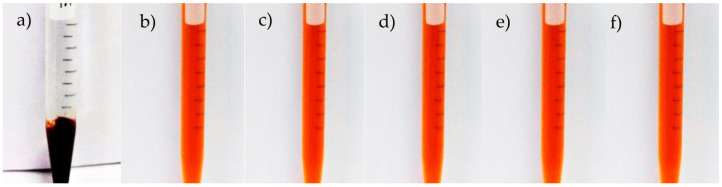
Visual confirmation of (**a**) unmodified iron hydroxide nanoparticles. (**b**) Fe:GA (1:0.25) (**c**) Fe:GA (1:0.5), (**d**) Fe:GA (1:1), (**e**) Fe:GA (1:1.5), and (**f**) Fe:GA (1:2).

**Figure 2 nanomaterials-11-01308-f002:**
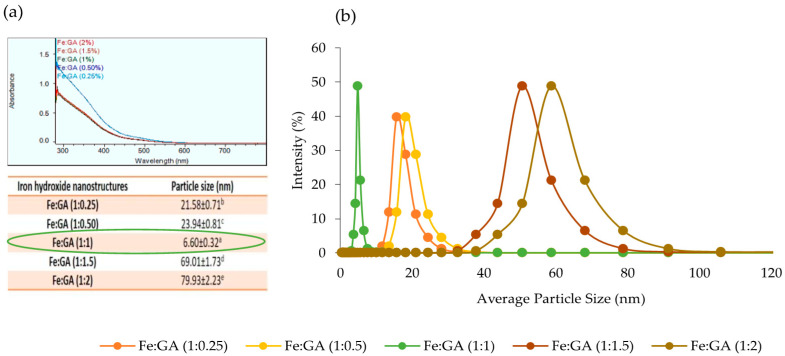
(**a**): Uv-Visible spectrum of IHNPs stabilized with 0.25, 0.50, 1.0, 1.5, and 2.0% gum arabic; (**b**): average particle size of gum arabic (0.25, 0.50, 1, 1.5, and 2.0%) capped IHNPs, Data are presented as means ± SEM (n = 3). ^a,b^ Means within the row with different lowercase superscript are significantly different (*p* < 0.05) from each other

**Figure 3 nanomaterials-11-01308-f003:**
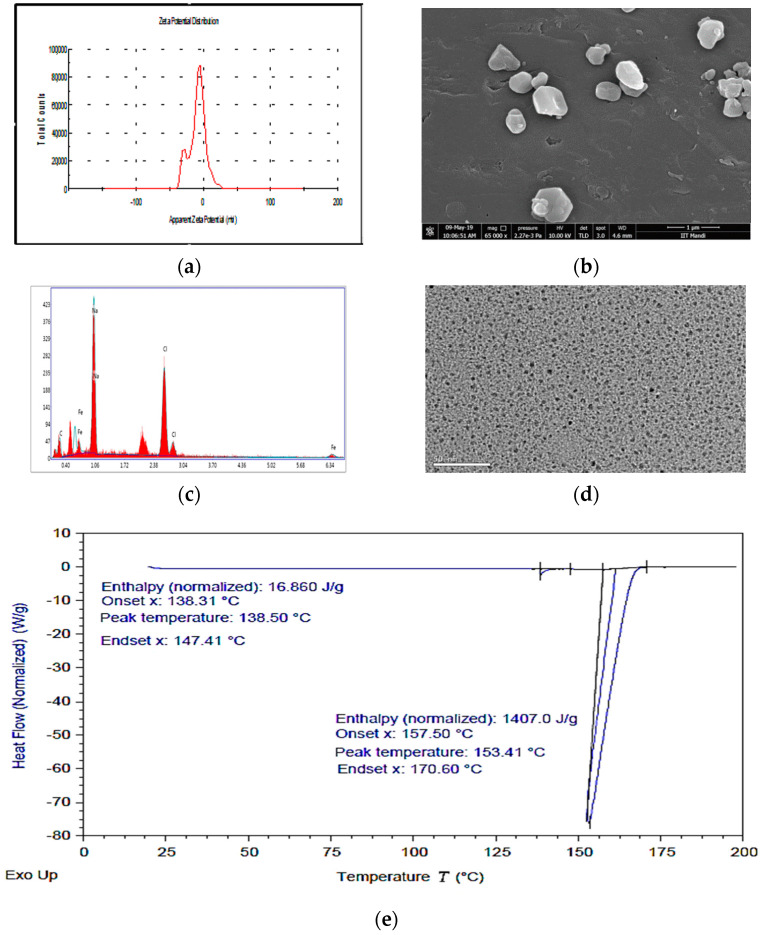
(**a**): apparent zeta potential of IHNPs, nanoparticles capped with 1% gum arabic; (**b**): Scanning electron microscopy image of IHNPs, capped with 1% gum arabic (2500×); (**c**): Energy-dispersive X-ray spectroscopy image of IHNPs, capped with 1% gum arabic; (**d**): Transmission electron microscopy image of IHNPs, capped with 1% gum arabic (50 nm); (**e**): Thermal stability of gum arabic stabilized IHNP’s, using a differential scanning technique.

**Figure 4 nanomaterials-11-01308-f004:**
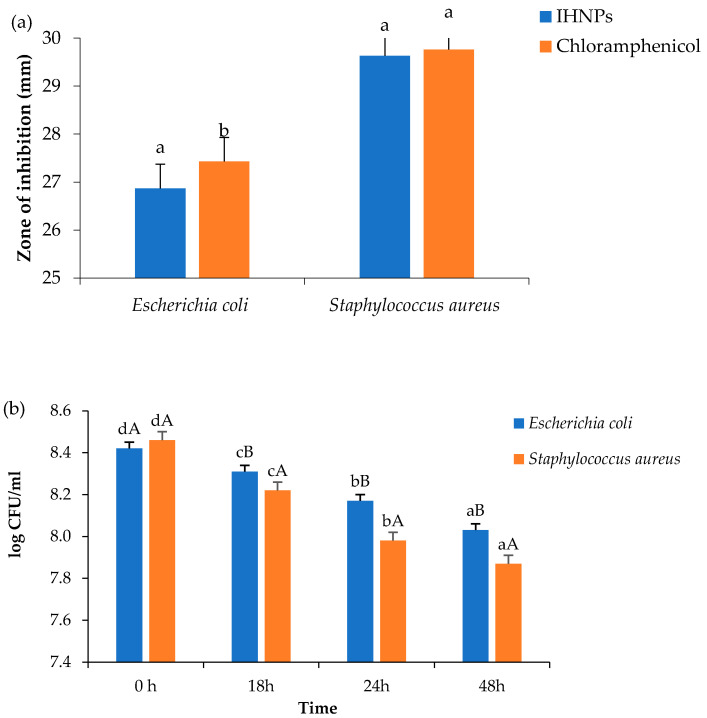
(**a**): Zone of inhibition; (**b**): Time kill study of gum arabic stabilized IHNP’s against pathogenic microorganisms. Data are presented as means ± SEM (n = 3). ^a,b^ Means within the row with different lowercase superscript are significantly different (*p* < 0.05) from each other; ^A,B^ Means within the row with different uppercase superscript are significantly different (*p* < 0.05) from each other.

**Figure 5 nanomaterials-11-01308-f005:**
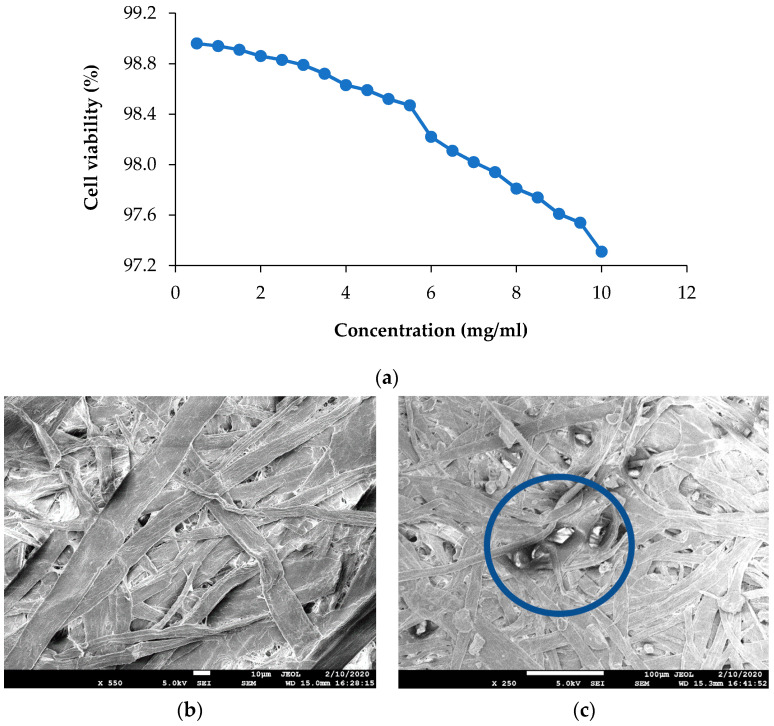
(**a**): Viability of Caco-2 cells exposed to increasing concentration of 1% gum arabic capped; IHNP’s; (**b**): Scanning electron microscopy image of control cellulose paper; (**c**): Scanning electron microscopy image of IHNPs, embedded cellulose paper.

**Figure 6 nanomaterials-11-01308-f006:**
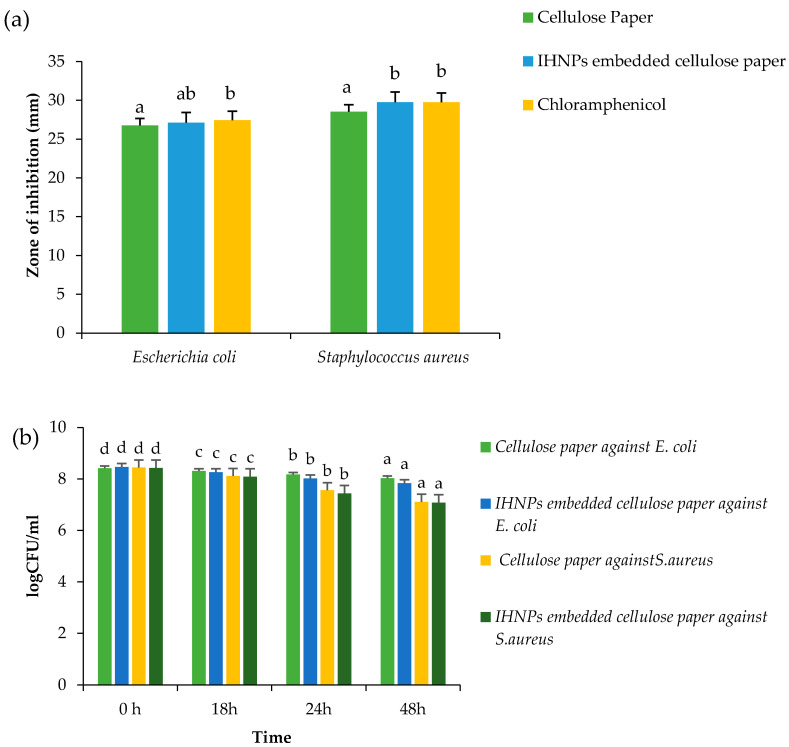
(**a**) Zone of inhibition (**b**) Time kill study of gum arabic stabilized IHNP’s embedded cellulose paper against pathogenic microorganisms. Data are presented as means±SEM (n = 3). ^a,b^ Means within the row with different lowercase superscript are significantly different (*p* < 0.05) from each other; ^a–d^ Means within the row with different lowercase superscript are significantly different (*p* < 0.05) from each other.

**Table 1 nanomaterials-11-01308-t001:** Shelf-life study of cellulose paper-wrapped paneer.

Microbiological AnalYsis.	Unwrapped Paneer (CFU)	Cellulose Paper-Wrapped Paneer (CFU)	IHNPs Embedded Cellulose Paper-Wrapped Paneer (CFU)
**0 day**			
Standard Plate Count	Not detected	Not detected	Not detected
Coliform Count	Not detected	Not detected	Not detected
Yeast and Mold Count	Not detected	Not detected	Not detected
**2nd day**			
Standard Plate Count	17 × 10^4^/g	Not detected	Not detected
Coliform Count	10/g	Not detected	Not detected
Yeast and Mold Count	Not detected	Not detected	Not detected
**4th day**			
Standard Plate Count	38 × 10^4^/g	Not detected	Not detected
Coliform Count	22/g	Not detected	Not detected
Yeast and Mold Count	Not detected	Not detected	Not detected
**6th day**			
Standard Plate Count	57 × 10^4^/g	2 × 10^4^/g	1 × 10^4^/g
Coliform Count	58/g	9/g	Not detected
Yeast and Mold Count	44/g	18/g	Not detected
**8th day**			
Standard Plate Count	72 × 10^4^/g	10 × 10^4^/g	3 × 10^4^/g
Coliform Count	89/g	20/g	1/g
Yeast and Mold Count	57/g	28/g	5/g
**10th day**			
Standard Plate Count	84 × 10^4^/g	34 × 10^4^/g	14 × 10^4^/g
Coliform Count	110/g	48/g	7/g
Yeast and Mold Count	69/g	49/g	12/g

## Data Availability

All data generated or analyzed during this study are included in this manuscript.

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
