# Peer review of "Potential of Gum Arabic Functionalized Iron Hydroxide Nanoparticles Embedded Cellulose Paper for Packaging of Paneer"

_nanomaterials, 2021, doi:10.3390/nano11051308_

Round 1
Reviewer 1 Report
The manuscript is scientifically sound and all methods and experiments are clearly presented and analyzed. The work is very well documented and very cohesive. Just some minor corrections are required before it can be published:
- The abstract should be re-written in a more appropriate way summarizing the scope and significance of the study, an outline of the experimental work and the major conclusions. In its present form, it contains very specific details of the experiments which are not appropriate for an abstract and in general it does not do justice to the work contained in the manuscript.
- Figure 2.e is of not good quality and it is suggested that it is removed or improved
- Figure 5c is missing its caption
Author Response
Answers to Reviewer 1:
Dear Reviewer,
Dear reviewer, the authors thank you for the opinion about our manuscript, which you provided in the introduction to the review. Thank you very much for reading our article and for valuable comments that are to improve our manuscript. All your comments have been included, changes made or further clarifications. To facilitate the review, we applied a red font to all changes from the original application.
In reference to the Reviewer comments, the following changes or further explanations in the article have been made:
Com. 1: The manuscript is scientifically sound and all methods and experiments are clearly presented and analyzed. The work is very well documented and very cohesive. Just some minor corrections are required before it can be published:
Ans. 2: Thank you so much for your encouraging comments on our manuscript. These comments always boost the confidence of the authors.
Com. 2. The abstract should be re-written in a more appropriate way summarizing the scope and significance of the study, an outline of the experimental work, and the major conclusions. In its present form, it contains very specific details of the experiments which are not appropriate for an abstract and in general it does not do justice to the work contained in the manuscript.
Ans. 2: Thanks for your suggestion. Abstract now has been modified and rewritten with the scope and significance of the study. Following lines have been added to the manuscript “Recently, scientists are interestingly turning to the eco-friendly metal nanoparticles due to their distinctive physicochemical properties that have been used in several biochemical and food applications including drug and bioactive component delivery, sensing of food pathogenic bacteria, imaging techniques, and theranostics, therefore this study was aimed to fabricate gum arabic stabilized iron hydroxide nanoparticles (IHNPs) using the co-precipitation process and to develop nanoparticles decorated antimicrobial cellulose paper. Agglomeration of IHNPs is a major concern, therefore, the varied concentration (0.25-2.0%) of gum arabic was used to functionalized and stabilized the nanoparticles, and based on UV-visible spectroscopy and particle size analysis, 1% gum arabic concentration was screened out. Scanning electron microscopy displayed polygonal disc shapes of IHNPs that had sides of approximately equal lengths. Energy dispersive spectroscopy was used to determine the purity of the IHNPs and results illustrated the elemental iron peak at 0.8keV and 6.34keV. For thermal stability, differential scanning calorimetry (DSC) was employed” (Line 15-26).
Com. 3: Figure 2.e is of not good quality and it is suggested that it is removed or improved.
Ans. 3: Thanks for pointing anomaly. Figure 2e now has been improved accordingly (Line 308-309).
Com. 4: Figure 5c is missing its caption
Ans. 4: Thanks for pointing anomaly. Caption for figure 5c now has been included in the manuscript (Line 397-398).
Thank you very much for the comments indicated in the manuscript, all of them have been included by making corrections in the text.
Reviewer 2 Report
The study was aimed to fabricate gum arabic stabilized iron hydroxide nanoparticles (IHNPs) using the co-precipitation process and developing nanoparticles decorated antimicrobial cellulose paper. The author prepared and verified the synthesis of IHNP. Besides, antimicrobial properties and cytotoxicity of IHNP also were evaluated, which showed certain potential in food application. The manuscript is of general interest to the readers of present journal. Nevertheless, the manuscript also has some explanations and mistakes needing to be addressed.
Some specific concerns and suggestion are as follows:
- In the title, the reviewerthink “food application” was inappropriately used because the author only explored the application in dairy product (Paneer).
- In line 109, whether IHNs was washed? What was chosen as the washing reagent?
- Why the concentration of 108 cells/mL waschosen for the antimicrobial properties of IHNPs in line 156? How much the IHNPs was taken?
- Why the chloramphenicol was used as a standard antibacterial agent in line 161? Was DMSO used as the solvent for IHNPs? Please explain to make it clear.
- In Figure 2a, did the ratio and percentage of Fe:GA mean the same thing?
- In Figure 2b, the legends should be placed in corresponding figure.
- In line 285, why were there sodium and chlorine in IHNPs? Please explain.
- In section 3.3, why did the antimicrobial effect of chloramphenicol was only carried out but no other antibiotics?
- In section 3.4, please explain the reason that Caco-2 and MB-231 cells were used to evaluate the cell viability. Besides, there was no detailed description that how to measure the cell viability (MTT assay).
- There has a space between numbers and unit. For example, in line 117, 118, 150 and so on.
- The italics was used to the name of bacteria. For example, in Figure 4b, Figure 6b.
- The abbreviation should show its full name, for example, EDS in line 282, DSC in line 290 and FESEM in line 373. Besides, the full name of FSSAI in line 431 should be placed in line 428.
- In line 246, “Figure 1h” should be “Figure 1b”.
- In Table 1, what does “ND” represent? Please explain clearly.
- In line 340, “coli” should be “E. coli”.
- In Figure 4b, the “Time” should be placed below the picture.
Author Response
Answers to Reviewer 2:
Dear Reviewer,
Dear reviewer, the authors thank you for the opinion about our manuscript, which you provided in the introduction to the review. Thank you very much for reading our article and for valuable comments that are to improve our manuscript. All your comments have been included, changes made or further clarifications. To facilitate the review, we applied a red font to all changes from the original application.
In reference to the Reviewer comments, the following changes or further explanations in the article have been made:
The study was aimed to fabricate gum arabic stabilized iron hydroxide nanoparticles (IHNPs) using the co-precipitation process and developing nanoparticles decorated antimicrobial cellulose paper. The author prepared and verified the synthesis of IHNP. Besides, antimicrobial properties and cytotoxicity of IHNP also were evaluated, which showed certain potential in food application. The manuscript is of general interest to the readers of the present journal. Nevertheless, the manuscript also has some explanations and mistakes needing to be addressed.
Thanks for your valuable suggestions and encouraging comments on our manuscript.
Some specific concerns and suggestion are as follows:
Com. 1: In the title, the reviewer thinks “food application” was inappropriately used because the author only explored the application in the dairy product (Paneer).
Ans. 1: Thanks for your valuable suggestion. Now the title of the manuscript has been revised to “Potential of gum arabic functionalized iron hydroxide nanoparticles embedded cellulose paper for packaging of Paneer” (Line 1-3).
Com. 2: In line 109, whether IHNPs was washed? What was chosen as the washing reagent?
Ans. 2: Thanks for pointing anomaly. The line has been included in the manuscript “IHNPs were then washed with triple distilled water, collected on a glass petri plate, and dried in a hot air oven at 45 ℃ for 4h” (Line 117-118).
Com. 3: Why the concentration of 108 cells/mL was chosen for the antimicrobial properties of IHNPs in line 156? How much the IHNPs was taken?
Ans. 3: According to the National Committee for Clinical Laboratory Standards (NCCLS) guidelines address quality control issues and current methodologies such as conventional disk diffusion methods (1×108 cells/ml) should be used. A 10 ml stock solution of different concentrations (25, 50, 75, 100 µl/ml) of IHNPs was prepared and a fixed volume (100 μl/well) from each concentration was poured into the prepared wells (Line 166).
Com. 4: Why the chloramphenicol was used as a standard antibacterial agent in line 161? Was DMSO used as the solvent for IHNPs? Please explain to make it clear.
Ans. 4: Thanks for pointing anomaly. To compare the antimicrobial activity of gum arabic stabilized IHNPs with a commercially available antibiotic component, chloramphenicol was used as a standard (positive control) antibacterial agent, whereas, DMSO was used as a negative control (Line 167-170).
Com. 5: In Figure 2a, did the ratio and percentage of Fe:GA mean the same thing?
Ans. 5: Yes, it is the same thing and it is already mentioned in the manuscript (Line 120; 250-258).
Com. 6: In Figure 2b, the legends should be placed in the corresponding figure.
Ans. 6: Thanks for your remark. Figure 2b is the combination of two results, hence, it is explained one legend as it is considered as one figure only.
Com. 7: In line 285, why were there sodium and chlorine in IHNPs? Please explain.
Ans. 7: Thanks for your remark. The sentence now has been rephrased accordingly. EDS spectrum illustrated the elemental iron peak at 0.8keV and 6.34keV. The presence of other elemental components such as carbon (0.3keV), sodium (1.06keV), and chlorine (2.8keV and 3.0keV) was confirmed, hence, the EDS spectrum was free of impurities confirming the purity of the IHNPs. Another elemental composition was attributed due to the utilization of NaOH during the synthesis of IHNPs (Line 291-295).
Com. 8: In section 3.3, why did the antimicrobial effect of chloramphenicol was only carried out but no other antibiotics?
Ans. 8: Chloramphenicol is a third-generation semisynthetic, broad-spectrum antibiotic derived from Streptomyces venequelae with primarily bacteriostatic activity. To compare the antimicrobial activity of gum arabic stabilized IHNPs with a commercially available antibiotic component, chloramphenicol was used as a standard (positive control) antibacterial agent. In this experiment, we evaluated the antimicrobial activity of IHNPs.
Com. 9: In section 3.4, please explain the reason that Caco-2 and MB-231 cells were used to evaluate the cell viability. Besides, there was no detailed description that how to measure the cell viability (MTT assay).
Ans. 9: Thanks for your remark. We have used only Caco-2 cells to determine the cytotoxicity of IHNPs.
It has already been mentioned in the following sentence Cytotoxicity of IHNPs was evaluated and the results of cell viability by MTT assay are represented in (Figure 5a). Herein, IHNPs were found to be non-toxic to Caco-2 cells. Higher concentration (10 mg/ml) of IHNPs showed 97.31% cell viability. Our results were correlated well with the literature [31] who studied the cytotoxic effect of albumin-coated copper and gum arabic caped copper nanoparticles on human breast cancer cells of MDA-MB- 231and Caco-2 cells and reported almost no toxicity of copper nanoparticles even at higher concentration.
Com. 10: There has a space between numbers and unit. For example, in line 117, 118, 150 and so on.
Ans. 10: Thanks for pointing anomaly. Now the space between numbers and unit has been removed throughout the manuscript.
Com. 11: The italics was used to the name of bacteria. For example, in Figure 4b, Figure 6b.
Ans. 11: Thanks for pointing anomaly. Changes now have been made as per the suggestion (Line 360 and 422).
Com. 12: The abbreviation should show its full name, for example, EDS in line 282, DSC in line 290 and FESEM in line 373. Besides, the full name of FSSAI in line 431 should be placed in line 428.
Ans. 12: Thanks for pointing anomaly. In the materials method section full name of EDS, DSc, and FESEM description is already mentioned, therefore we used abbreviations in further sections. Full form Food Safety Standard Authority of India (FSSAI) now has been added to the manuscript (Line 444).
Com. 13: In line 246, “Figure 1h” should be “Figure 1b”.
Ans. 13: Thanks for pointing anomaly. Changes made accordingly (Line 254).
Com. 14: In Table 1, what does “ND” represent? Please explain clearly.
Ans. 14: Thanks for pointing anomaly. ND now changed with ‘Not detected’ throughout the table (Line 446-447).
Com. 15: In line 340, “coli” should be “E. coli”.
Ans:. 15 Thanks for pointing anomaly. Changes made accordingly (Line 349).
Com. 16: In Figure 4b, the “Time” should be placed below the picture.
Ans. 16: Thanks for pointing anomaly. Changes made accordingly (Line 360-361).
Thank you very much for the comments indicated in the manuscript, all of them have been included by making corrections in the text.

Round 2
Reviewer 2 Report
My concerns have been addressed, the manuscript is now acceptable.